# Transcriptome and Metabolite Conjoint Analysis Reveals the Seed Dormancy Release Process in Callery Pear

**DOI:** 10.3390/ijms23042186

**Published:** 2022-02-16

**Authors:** Jing Zhang, Jia-Yi Qian, Yue-Hong Bian, Xiao Liu, Chun-Lei Wang

**Affiliations:** School of Horticulture and Plant Protection, International Research Laboratory of Agriculture and Agri-Product Safety, Key Laboratory of Plant Functional Genomics of the Ministry of Education, Yangzhou University, 48 Wenhui East Road, Yangzhou 225009, China; zhangj45@yzu.edu.cn (J.Z.); qianjy021@163.com (J.-Y.Q.); byh18352766210@163.com (Y.-H.B.); liuxiao@yzu.edu.cn (X.L.)

**Keywords:** transcriptome, cold stratification, hormone, carbohydrate metabolism, seed dormancy, dormancy release

## Abstract

Seed dormancy transition is a vital developmental process for seedling propagation and agricultural production. The process is precisely regulated by diverse endogenous genetic factors and environmental cues. Callery pear (*Pyrus calleryana* Decne) is an important rootstock species that requires cold stratification to break seed dormancy, but the mechanisms underlying pear seed dormancy release are not yet fully understood. Here, we analyzed the transcriptome profiles at three different stages of cold stratification in callery pear seeds using RNA sequencing combined with phytohormone and sugar content measurements. Significant alterations in hormone contents and carbohydrate metabolism were observed and reflected the dormancy status of the seeds. The expressions of genes related to plant hormone metabolism and signaling transduction, including indole-3-acetic acid (IAA) biosynthesis (*ASAs*, *TSA*, *NITs*, *YUC*, and *AAO*) genes as well as several abscisic acid (ABA) and gibberellic acid (GA) catabolism and signaling transduction genes (*CYP707A*s, *GA2ox*, and *DELLA*s), were consistent with endogenous hormone changes. We further found that several genes involved in cytokinin (CTK), ethylene (ETH), brassionolide (BR), and jasmonic acid (JA) metabolism and signaling transduction were differentially expressed and integrated in pear seed dormancy release. In accordance with changes in starch and soluble sugar contents, the genes associated with starch and sucrose metabolism were significantly up-regulated during seed dormancy release progression. Furthermore, the expression levels of genes involved in lipid metabolism pathways were also up-regulated. Finally, 447 transcription factor (TF) genes (including *ERF*, *bHLH*, *bZIP*, *NAC*, *WRKY*, and *MYB* genes) were observed to be differentially expressed during seed cold stratification and might relate to pear seed dormancy release. Our results suggest that the mechanism underlying pear seed dormancy release is a complex, transcriptionally regulated process involving hormones, sugars, lipids, and TFs.

## 1. Introduction

Pear (*Pyrus* spp.) is one of the most widely cultivated fruit crops owing to its strong adaptability. As perennial woody plants, pear species evolved dormancy (ability to time the germination) to avoid injury in unsuitable environments, and this trait occurs in both seeds and buds. Seed dormancy inhibits seeds from germinating at the wrong time or in the wrong space, whereas bud dormancy ensures that vegetative growth occurs only under optimal environmental conditions [1]. During the production of pear nursery stock, grafting is used to produce pear seedlings. Specifically, the seed of rootstock cultivars is planted first and then scion cultivars are grafted to the rootstock seedlings. Hence, high-quality rootstock seed with rapid and uniform germination is a prerequisite for pear stock production. However, the germination ability of rootstock seeds cannot be assessed for their dormancy property [2]. Pear seeds have characteristics of non-deep physiological and physical dormancy, i.e., combinatorial dormancy, with dormancy caused by both embryo physiology and water-impermeable seed coats [3,4]. Whether the dormancy of seeds has been released is a matter directly related to seedling establishment and affects the production of pear nursery stock. Therefore, elucidating the mechanism of pear seed dormancy release will be useful in the cultivation of pear seedlings.

Temperature is the main environmental cue that alters pear seed dormancy breaking [2,5]. Moist chilling (referred to as cold stratification) is broadly applied to break pear seed dormancy and promote the breeding cycle [6]. Meanwhile, some strategies, such as seed-coat removal and chemical treatment of seeds, have also been developed to shorten the dormancy release progression and improve the germination rate in pear seeds [5,7].

Induction, maintenance, and release of dormancy are consecutive processes in seed that are determined by many physiological processes, especially the regulation of phytohormones [8,9]. Abscisic acid (ABA) and gibberellic acid (GA) are the primary hormone mediators in regulating seed dormancy and germination, and they act in an antagonistic manner. ABA triggers the establishment of dormancy, whereas GA alleviates dormancy and promotes seed dormancy release and germination. [10,11]. Modulating the expression of a series of genes involved in ABA and GA biosynthesis, catabolism, and signal transduction results in altered seed dormancy and germination [11]. Meanwhile, the transcription factors (TFs) *ABA INSENSITIVE3* (ABI3), *ABI4*, and *ABI5* also play important roles in seed dormancy and germination regulation, primarily by targeting genes related to ABA and GA biosynthesis, catabolism, and signal transduction [12,13,14,15]. In pear seeds, both the seed coat and the embryo contain high concentrations of ABA during the induction of dormancy and the concentration of ABA is reported to decrease during cold stratification whereas that of GA3 is reported to increase, thereby initiating dormancy release [4,5]. The expression levels of the *ABI* genes are significantly differentially expressed in pear seeds with distinct dormancy stages, which might participate in dormancy regulation of pear seeds as well [4].

It has been reported that the seed reserve content is closely related to seed dormancy and the germination process [16,17]. Starch and soluble sugars (including sucrose, glucose, and fructose) are important carbohydrate reserves in plant seeds that not only provide essential energy sources but also act as a signal molecule in diverse physiological processes [18]. Sucrose may be converted to starch for longer-term storage during seed maturation, and the storage starch is remobilized to support seed dormancy release, germination, and subsequent seedling establishment [19]. Additionally, enzymes and corresponding genes associated with starch–sucrose conversion have been widely studied in many species [20,21]. Emerging findings also report that sucrose and starch are hydrolyzed into glucose by sucrose synthase (SUS/SS) and α-amylase (AMY) during pear seed dormancy release and the inhibition of *PcaSS8* blocks pear seeds from germinating [22].

Breaking of seed dormancy is controlled by various regulators with implications for physiological processes, including hormone levels, energy metabolism, mobilization of storage material, and environmental stimuli [23]. Despite the tremendous breakthrough in dormancy release regulatory networks in seeds of Arabidopsis and cereals, relatively few investigations have attempted to characterize the mechanism concerning the dormancy release of pear seeds. To explore the dormancy release mechanism of pear seeds, callery pear (*Pyrus calleryana* Decne) seeds were selected and treated with cold stratification. We further separated seed dormancy release progression into three stages according to the seed dormancy level. A global transcriptional analysis was conducted using RNA sequencing and helped identify numerous differentially expressed genes at three different dormant stages. Additionally, the hormone and sugar contents were measured using high-performance liquid chromatography. These results offer comprehensive expression profile data for in-depth characterization of crucial genes during dormancy release progression in pear seeds.

## 2. Results

### 2.1. Evaluation of the Seed Dormancy Status and Phytohormones in Pear Seed during Dormancy Release Progression

Quantification of the seed dormancy status is a prerequisite of dormancy studies. In this study, a germination experiment was performed to estimate the seed dormancy level. The germination percentages of seeds sampled at discrete periods of cold stratification and cultured at a constant temperature of 25 °C under 16 h daylight in the growth chamber for 10 days are shown in Figure 1A. The mature imbibed seeds (0 days of cold stratification) were kept in deep dormancy and severely inhibited from germinating. The germination rate was 20% after 20 days of cold stratification. After 40 days of cold stratification, the seed germination rate was rapidly increased and reached 85% after 50 days of cold stratification, indicating that the chilling requirement is satisfied and the seeds completely break dormancy (Figure 1A). As the seeds after 0, 20, and 50 days of cold stratification exhibited distinct germination rates, we further analyzed the phytohormone contents at the three dormant times (Figure 1B–E). The ABA content declined sharply from day 0 to day 20 (by 72.1%), but no significant change was observed from day 20 to day 50 (Figure 1B). The IAA content gradually increased during seed dormancy release progression, with 43.1% increment from day 0 to day 20 and 92.6% increment from day 20 to day 50 (Figure 1C); interestingly, a similar trend was also observed for the GA3 (an important active gibberellin) content. The GA3 content (11.8 ± 2.4 ng/g FW) was low on day 0 and then steady climbed from day 0 to day 50, reaching 21.4 ± 3.6 ng/g FW (Figure 1D). The ABA/GA3 ratio was observed with a significant decrease from day 0 to day 20 (by 80.8%) but with little change between day 0 and day 50 (Figure 1E).

### 2.2. Carbohydrate Contents in Seeds during Dormancy Release Progression

Carbohydrate metabolism provides energy and plays a vital role in dormancy release regulation, and it is widely reported that the stored starch converts into soluble sugars during seed cold stratification [24]. Thus, the contents of starch and soluble sugars were analyzed at the three dormant times (Figure 2). The starch content was high in imbibition seeds (day 0) and then decreased significantly (by 59.2%) from day 0 to day 50 (Figure 2A). The level of sucrose was relatively constant during stratification (Figure 2B). The contents of glucose and fructose underwent consistent changing trends, significantly increasing from day 20 to day 50 and reaching 2.32 and 7.39 mg/g FW, respectively (Figure 2C,D).

### 2.3. Transcriptome Analysis of Seeds during Dormancy Release Progression

To gain further insight into the transcriptome changes in the seed dormancy release progression of callery pear, seeds sampled at different dormant times (0, 20, and 50 days of cold stratification) were subjected to RNA sequencing analysis. After stringent quality filtering and trimming, approximately 132,008,082; 131,375,566; and 123,484,610 clean reads with a Q30 percentage of 93% were identified from the three dormancy stages (Appendix A). The reads were then mapped to the reference genome sequence of wild pear *P. betuleafolia*. The aligned ratio of the reads ranged from 79.12 to 89.14% among the nine libraries, and 75.32–85.1% of the reads were uniquely mapped (Appendix A).

Based on the pairwise comparison of the absolute value of the log_2_ (fold change) with FPKM ≥1 and a correlated *p*-value ≤ 0.01 as the threshold, we ultimately identified 8048 DEGs during seed dormancy release (Appendix A). Among these, 4077 (2035 up- and 2042 down-regulated), 842 (284 up- and 558 down-regulated), and 7141 (3898 up- and 3243 down-regulated) DEGs were significantly enriched in 0 days versus 20 days, in 20 days versus 50 days, and in 0 days versus 50 days, respectively (Figure 3A). A Venn diagram analysis was also conducted to uncover the overlap between different comparison groups (Figure 3B). A total of 4240 (52.7%) genes exhibited stage-specific expression patterns, but only a small fraction of transcripts (204, 2.5%) was found in all comparison groups. These common differentially expressed genes may play key roles in regulating the seed dormancy release.

To further characterize the expression profiles, the entire 8048 DEGs were hierarchical clustered into eight clusters with distinct expression patterns and each cluster was examined by KEGG pathway enrichment (Figure 3C). The largest cluster, Cluster 1 (2095 DEGs), was predominantly expressed in imbibed mature seeds and highly associated with basal TFs. Cluster 2, consisting of 1865 DEGs, was involved in SNARE interactions in vesicular transport, protein processing in endoplasmic reticulum, ubiquitin-mediated proteolysis, and autophagy (other eukaryotes). Cluster 3 (273 DEGs) was mainly enriched in genes encoding flavonoid biosynthesis, photosynthesis, and glutathione metabolism. DEGs of Cluster 4 were mainly enriched in ribosome. Clusters 5 and 6 displayed expression patterns opposite to those of Clusters 3 and 4, respectively, and were not significantly enriched in biological processes or pathways. In Cluster 7, 1340 DEGs displayed an increasing trend and were overrepresented in items such as carbon fixation in photosynthetic organisms, glycolysis/gluconeogenesis, and fructose and mannose metabolism. Cluster 8, with the second-largest number of DEGs, was significantly enriched in pathways such as pyruvate metabolism; alanine, aspartate, and glutamate metabolism; and fatty acid degradation. These results provide important cues for speculating on mechanisms involved in seed dormancy release regulation.

### 2.4. DEGs Involved in the Biosynthesis, Metabolism, and Signaling Transduction of Plant Hormones during Seed Dormancy Release Progression

A variety of plant hormones constitute a complex dormancy regulation network. Genes involved in plant hormone synthesis, metabolism, and signaling, including those associated with ABA, GA, IAA, cytokinin (CTK), ethylene (ETH), brassionolide (BR), and jasmonic acid (JA), were significantly expressed during seed dormancy release progression (Appendix A).

Nine-cis-epoxycarotenoid dioxygenase (NCED) and short-chain alcohol dehydrogenase (SDR) are two key enzymes regulating ABA synthesis. As shown in Figure 4, the expression levels of genes encoding SDR were down-regulated in the dormancy release stages and were consistent with the reduction in ABA contents whereas genes encoding NCED were not correlated with ABA concentrations. Correspondingly, three DEGs encoding ABA 8′-hydroxylase (*CYP707A*), which are involved in ABA catabolism, gradually increased during seed dormancy release and showed an opposite pattern to that of ABA contents (Figure 4A and Appendix A). The genes encoding ABA receptors (*PYL*) and protein phosphatase 2C (*PP2C*), which are involved in ABA signaling transduction, were also differentially expressed during these times (Figure 4B and Appendix A).

In the GA biosynthesis pathway, genes encoding ent-kaurene synthase (*KS*) and ent-kaurene oxidase (*KO*) were increased from day 0 to day 20 and then decreased from day 20 to day 50 but the DEG encoding gibberellin 20 oxidase (*GA20ox*) was down-regulated in the dormancy release progression. Meanwhile, two DEGs encoding ent-kaurenoic acid oxidase (*KAO*) showed opposite expression profiles (Figure 4A and Appendix A). Additionally, the DEGs encoding gibberellin 2 oxidase (*GA2ox*) that participated in GA inactivation were gradually decreased during these stages, with *Chr10.g15067* the most abundantly expressed (Appendix A). Our data shows that the expression levels of several genes related to GA signaling and response, such as genes encoding DELLA proteins and F-box proteins SLEEPY (*SYL*), were significantly up-regulated during seed cold stratification but genes encoding scarecrow-like proteins (*SCL*) and gibberellin-regulated proteins (*GAST*) showed diverse expression patterns during seed cold stratification (Figure 4B and Appendix A).

The following IAA synthesis pathway genes as well as the IAA levels were significantly activated during seed cold stratification: two alpha subunits of anthranilate synthase (*ASA*) genes, one tryptophan synthase alpha (*TSA*) gene, one indole-3-pyruvate monooxygenase (*YUCCA*) gene, two nitrolase (*NIT*) genes, and one aldehyde oxidase (*AAO*) gene (Figure 4A and Appendix A). However, the DEGs encoding transport-inhibitor-resistant 1/auxin signaling F-box (*TIR1*/*AFB*) proteins, indole-acetic acid (*IAA*) proteins, Gretchen Hagen3 (*GH3*), and small auxin-up RNAs (*SAUR*), which are involved in IAA signaling transduction and response, showed distinct expression patterns during the dormancy release progression of callery pear seeds (Figure 4B and Appendix A).

The expression levels of genes encoding LONELY GUY (*LOG*), involved in CTK synthesis; s-adenosylmethionine synthase (*SAMS*) and 1-aminocyclopropane-1-carboxylate synthase (*ACS*), involved in ETH synthesis, were gradually decreased during dormancy release progression, while three DEGs encoding 1-aminocyclopropane-1-carboxylate oxidase (*ACO*), involved in ETH synthesis, were gradually increased during the three dormant times (Figure 4A and Appendix A). Additionally, several DEGs associated with CTK and ETH signaling transduction, such as histidine-containing phosphor-transfer protein (*AHP*), two two-component response regulators (*ARR*), reversion-to-ethylene sensitivity (*RTE*) protein, and ethylene-insensitive protein 3 (*EIN3*), were significantly down-regulated in pear seeds during cold stratification (Figure 4B and Appendix A).

We also analyzed the expression levels of genes associated with BR and JA biosynthesis and signaling transduction and identified a series of genes, including those encoding cytochrome P450 enzyme (*CYP90A1*), brassinosteroid insensitive1 (*BRI*), GSK3-like kinase brassinosteroid insensitive2 (*BIN*), lipoxygenase (*LOX*), 12-oxophytodienoate reductase (*OPR*), OPC8:0-CoA ligase (*OPCL*), jasmonic acid amido synthetase (*JAR*), and jasmonate ZIM domain-containing protein (*JAZ*), that were differentially expressed during seed cold stratification (Figure 4 and Appendix A).

### 2.5. DEGs Involved in Carbohydrate Biosynthesis and Metabolism during Seed Dormancy Release Progression

During seed dormancy release progression, starch showed a significant downward trend while soluble sugars (including fructose and glucose) showed the opposite changes (Figure 2). Meanwhile, KEGG enrichment analysis showed that the DEGs in Cluster 7 and Cluster 8 were significantly enriched in carbohydrate biosynthesis and metabolism pathways, including glycolysis/gluconeogenesis, amino sugar and nucleotide sugar metabolism, and pyruvate metabolism (Figure 3). Thus, we investigated the regulation of starch and sucrose metabolism pathways and identified 54 genes that were significantly differentially expressed during seed cold stratification (Figure 5 and Appendix A). The expression levels of DEGs encoding invertase (*INV*), invertase inhibitor (*Invlnh*), *SUS*, hexokinase (*HXK*), fructokinase (*FRK*), and glucose-6-phosphate isomerase (*PGI*) were significantly up-regulated in seeds after 20 and 50 days of cold stratification and accelerated the deposition of fructose and glucose during seed dormancy release progression (Figure 5 and Appendix A). Meanwhile, the DEGs associated with starch synthesis, including ADP-glucose pyrophosphorylase (*AGPase*), starch synthase (*SS*), granule-bound starch synthase (*GBSS*), and starch branching enzyme (*SBE*), were still gradually increased during the three dormant times (Figure 5 and Appendix A). AMY, β-amylase (BAM), glucan water dikinase (GWD), and isoamylase (ISA) are critical enzymes that catalyze starch degradation [19]. In this study, two *AMY* encoding genes, one *BAM* encoding gene, two *GWD* encoding genes, and one *ISA* encoding gene were identified with significant increasing trends and the fact that *Chr8.g54498* (an *AMY* encoding gene) is the most abundantly expressed (Appendix A) was also established. Meanwhile, the enzymes involved in interconversions between starch and sucrose, such as phosphoglucomutase (PGM), glucose-6-phosphate/phosphate translocator protein (GPT), and UDP-glucose pyrophosphorylase (UGPase), were identified with coding genes significantly up-regulated during seed dormancy release progression (Figure 5 and Appendix A). There were still some key genes in starch and sucrose metabolism that were differentially expressed, including one sucrose phosphate synthase (*SPS*) gene, four phosphofructokinase (*PFK*) genes, two sucrose transport protein (*SUT*) genes, and three sorbitol dehydrogenase (*SDH*) genes. Most of the DEGs were up-regulated during the three dormant times (Figure 5 and Appendix A).

### 2.6. DEGs Involved in Lipid Synthesis and Catabolism during Seed Dormancy Release Progression

Several genes involved in lipid synthesis and metabolism were significantly differentially expressed during seed cold stratification (Figure 6 and Appendix A). The expression levels of three glycerol-3-phosphate acyltransferase (*GPAT*) genes, one phosphatidic acid phosphatase (*PAP*) gene, and three phospholipid:diacylglycerol acyltransferase (*PDAT*) genes involved in triacylglycerol (TAG) synthesis were up-regulated during seed cold stratification (Figure 6 and Appendix A). Regarding lipid catabolism pathways, the DEGs encoding TAG lipase involved in oil breakdown, long-chain acyl-CoA synthase (*LACS*), acyl-CoA oxidase (*ACX*), multifunctional protein (*MFP*) and 3-ketoacyl-CoA thiolase (*KAT*) involved in β-oxidation, malate synthase (*MLS*), glyoxysomal malate dehydrogenase (*MDH*), and citrate synthase (*CSY*) and isocitrate lyase (*ICL*) involved in glyoxylate cycle were identified with a significant increasing trend (2- to 53-fold) during seed dormancy release progression, as was the fact that *Chr6.g51605* (the *MLS* encoding gene) is the most abundantly expressed (Appendix A).

### 2.7. TFs Were Differentially Expressed in Callery Pear Seed during Dormancy Release Progression

Given the crucial role of TFs in regulating seed dormancy and germination, we analyzed the TF gene families that were differentially expressed at different dormant times and prepared a list (Appendix A). A total of 448 TFs distributed across 40 TF families were differentially expressed during seed dormancy release progression (Figure 7A and Appendix A). The TF families with the most identified DEGs were ERFs (51 genes, 11.4%), bHLHs (40 genes, 9.0%), bZIPs (38 genes, 8.5%), NACs (30 genes, 6.7%), WRKYs (27 genes, 6.0%), and MYBs (22 genes, 4.9%) (Figure 7A). Among them, most of the DEGs belonging to ERF (40 members), NAC (24 members), and WRKY (23 members) families exhibited a downward trend during the cold stratification of callery pear seeds. The remaining *ERF* genes (11 members), *NAC* genes (6 members), and *WRKY* genes (4 members) exhibited the opposite expression pattern (Figure 7B). Simultaneously, the expression levels of 22 *bHLH* genes, 15 *bZIP* genes, and 10 *MYB* genes were significantly up-regulated from day 0 to day 50 (Figure 7B). Among the differentially expressed TFs, we also identified the homologs of *DOG1* (*DELAY OF GERMINATION 1*), *ABI4* (an ERF gene), and *ABI5* (a bZIP gene), which play a vital role in versatile pathways regarding seed dormancy and germination, as well as the homolog of *ICE1*, a master regulator that modulates plant cold response [15,25] (Appendix A). From these results, it was speculated that various differentially expressed TFs act as potential regulators of biological processes responsible for the dormancy release progression of callery pear seeds.

### 2.8. Validation of RNA-Seq by qRT-PCR Analysis

To confirm the reliability of the DEGs obtained by RNA-seq, a total of 12 genes, including 8 structural genes (*CYP707A2*, *PP2C50*, *GA2ox1*, *JAZ8*, *ACS1*, *AGPase1*, *SBE1*, and *MLS*) and 4 TFs (*AGL24*, *ABI4*, *RAV2*, and *NAC42*), were selected for qRT-PCR analysis. As shown in Figure 8, most of the selected genes displayed high correlations between qRT-PCR and transcriptomic data sets. Thus, the expression profiles of DEGs obtained from RNA-seq data should be efficient and believable.

## 3. Discussion

Seed dormancy is an innate seed property that ensures the distribution of germination in time and space [26]. Pear seed dormancy is characterized as both physiological and physical dormancy. Cold stratification (low temperature and moisture conditions) has been widely applied to break seed dormancy and promote subsequent germination, with the optimum stratification period varying among seeds of different pear rootstock species, ranging from 30 to 70 days [2,4,27]. Callery pear is an important rootstock species of pear native to China, with intermediate chilling requirement to break seed dormancy, but the factors that release callery pear seed dormancy are hardly known at the molecular level. To draw a complete dormancy release cycle during callery pear seed cold stratification, we conducted a transcriptomic study at three critical developmental times and provided insight into the molecular mechanism of seed dormancy release in callery pear.

### 3.1. Metabolism and Regulation of Plant Hormones in Callery Pear Seed Dormancy Release Progression

The importance of hormone homeostasis in controlling seed dormancy and release has been extensively reviewed in the last decades [11,28]. ABA and GA are the major hormones that antagonistically regulate the seed dormancy status. In the present investigation, the ABA level significantly declined from day 0 to day 20 and the GA3 (an important active gibberellin) concentration steadily increased during seed cold stratification (Figure 1B,C). In addition, the absolute ABA content was much higher than the absolute GA3 content and the ABA/GA3 ratio showed a similar trend to ABA content (Figure 1B–E). These findings are consistent with those in the seeds of red bayberry [29], celery [30], and ginkgo [31]. Simultaneously, the expression patterns of several ABA and GA biosynthesis and deactivation genes, including *SDRs*, *CYP707A*s, *KO*, *KAO*, and *GA20oxs*, exhibited high correlations with the changes in ABA and GA3 levels (Figure 4A). Among these DEGs, *CYP707A*s encode ABA 8′-hydroxylase, a predominant enzyme catalyzing the hydrolyzation of ABA into phaseic acid and/or dihydrophaseic acid. Previous researchers have identified four members in the *Arabidopsis thaliana CYP707A* gene family (*CYP707A1* to *CP707A4*) that play pivotal roles in controlling the level of ABA [32]. In this study, three *CYP707A* members with increasing expression levels during seed cold stratification were homologous to *AtCYP707A1* and *AtCYP707A2* and were indispensable in controlling seed dormancy and germination, leading to the speculation that *PcaCYP707A*s may play important roles in regulating the seed dormancy release progression of callery pear (Figure 4A) [33,34]. In addition to ABA and GA biogenesis, their perception by receptors signaling transduction and regulation also affects seed dormancy release [35,36]. PP2C proteins are negative regulators of the ABA signaling pathway [37]. Until now, several of the group A PP2C proteins in Arabidopsis, including *ABI1*, *ABI2*, *HAB1*, *HAB2*, *AHG1*, *AHG3*, and *HONSU*, have been reported to be involved in seed dormancy regulation, where the over-expression or loss of function of the corresponding genes significantly altered seed dormancy [38,39,40,41]. *Chr8.g54174*, a homolog of *AtHAB1*, was expressed with an increasing trend during seed cold stratification, as confirmed by RNA-Seq and qRT-PCR, and might contribute to callery pear seed dormancy release (Figure 8 and Appendix A). The functions of the other differentiated PP2C proteins await further clarification (Figure 4B and Appendix A). GA acts through the GID1-DELLA-SCFSLY1/GID2 signaling cascade [42,43]. The present study identified four differentially expressed DELLA proteins exhibiting high expression levels upon 20–50 days of cold stratification of callery pear seeds (Figure 4B). Studies have shown that the major effect of GA is the degradation of the DELLA protein, and DELLA protein is a key negative regulator of the GA signaling pathway [44]. *GAI* (*GA-INSENSITIVE*), *RGA* (*REPRESSOR-OF-GA*), and *RGL* (*REPRESSOR-OF-GA LIKE*) belong to the DELLA protein subfamily. It has been reported that *Arabidopsis RGL2* negatively regulates seed germination and a mutant of *rgl2* promotes seed germination [45,46]. Lv et al. [47] also reported that *PmRGL2* plays a negative role in dormancy release by regulating the GA biosynthetic enzymes in the *Japanese apricot* leaf bud. These studies above provide evidence of the potential roles of the DEGs associated with ABA and GA in the seed dormancy release of callery pear.

IAA is an essential phytohormone in regulating seed dormancy and germination. Several observations have implied that IAA participates in cross talk with ABA and thereby promotes seed dormancy but inhibits seed germination and pre-harvest sprouting [48,49,50]. In terms of seed dormancy release, the IAA level was detected with an increasing trend during seed imbibition and after ripening in *Arabidopsis*, pea, and wheat [51,52,53]. Consistent with previous studies, pear seed IAA levels and the corresponding IAA biosynthetic genes were significantly induced during seed cold stratification (Figure 1C and Figure 4), which indicates that auxin may play a role in regulating the seed dormancy release of callery pear.

Other plant hormones, such as CTK, ETH, BR, and JA, were also implicated in regulating seed dormancy, primarily by mediating the ABA/GA balance [35,54]. CTK, ETH, and BR positively regulate seed dormancy release in dicot species, and they function by modulating ABA synthesis, catabolism, and signaling transduction [8,55,56]. Analysis of our transcriptomic data identified 28 DEGs annotated as CTK, ETH and BR metabolic and signaling genes, but they exhibited distinct expression patterns during seed cold stratification (Figure 4). Their functions in seed dormancy release need further investigation. As for JA, 18 differentially expressed JA synthesis, catabolism, and signaling genes were obtained from transcriptomic data and most of the DEGs exhibited down-regulated expression patterns during seed cold stratification, indicating that JA might negatively regulate the seed dormancy release of callery pear (Figure 4). The results are consistent with those of *Arabidopsis* and wheat seeds during imbibition and dormancy breaking progress [51,53].

### 3.2. Metabolism and Regulation of Storage Reserves in Callery Pear Seed Dormancy Release Progression

The transition from seed dormancy to germination has been associated with the degradation and mobilization of storage reserves (starch, oils, and proteins), which are essential energy sources and contribute to the generation of dormancy release and subsequent seed germination [17,57,58]. During seed dormancy release and germination, starch, oils, and proteins are degraded and converted into soluble sugars, fatty acids, and amino acids, respectively. The starch-soluble sugar metabolic switch was also observed in the present investigation. Starch, the main carbohydrate reserve in pear seeds, was significantly decreased during seed cold stratification; and soluble sugars (including fructose and glucose) showed an opposite dynamic pattern (Figure 2). In the meantime, large numbers of the enzyme genes related to starch degradation and sucrose metabolism pathways, including *AMY*, *BAM*, *GWD*, *ISA*, *PGM*, *GPT*, *UGPase*, *INV*, *Invlnh*, *SUS*, *HXK*, *FRK* and *PGI*, were significantly up-regulated during this process, in accord with the alteration in starch and soluble sugars (Figure 5). The genes involved in starch synthesis (*AGPase*, *SS*, *GBSS*, and *SBE*), as well as hexogenesis, (*PGM*, *GPT*, *UGPase*, *PGI*, *SPS*, and *PFK*), were also increased with the dormancy release process (Figure 5). In addition, oil (triacylglycerol or TAG) is still the main store of carbon and accounts for approximately 20% of a pear seed’s weight [59]. Use of storage oils commences locally during dormancy release and germination [17,60,61]. TAG is firstly broken down to free fatty acids (FAs) and glycerol. FAs are then fed to the β-oxidation and glyoxylate pathways, which can supply malate, succinate, and acetyl CoA to the TCA cycle [62]. Here, the transcriptomic data showed that several genes involved in TAG synthesis, β-oxidation, and glyoxylate cycle are significantly enriched with an increasing trend upon seeds’ cold stratification, indicating the need for more energy supply during seed dormancy release (Figure 6). Moreover, carbohydrate substrates synergetic with hormones are orchestrated to finely regulate the dormancy and/or germination of seeds. DELLA proteins could be one of the main convergent actors of the hormonal and carbohydrate-dependent molecular networks. Loss of function of DELLA proteins has been found to accelerate seed starch and FA breakdown by up-regulating *AMY* and *SFAR* genes, respectively, and promote seed dormancy release and germination [63,64]. In rice, exogenous glucose alleviated ABA degradation by inhibiting ABA catabolism genes *OsABA8ox2* and *OsABA8ox3* and then prolonged seed dormancy release progression [65].

### 3.3. Transcription Factors Play a Pivotal Role in Callery Pear Seed Dormancy Release Progression

We also identified a plethora of TF genes that are differentially expressed in the dormancy release process (e.g., *ERF*, *bHLH*, *bZIP*, *NAC*, *WRKY*, and *MYB* genes) (Figure 7). Our observations are consistent with those of previous studies, determining that cold stratification induces various physiology changes by activating many TFs, including *ERF*, *MYB*, *bHLH*, *NAC**, WRKY*, and *MADS* genes [1,35,66]. For example, the expression levels of 40 differentially expressed *ERF* genes were down-regulated during seed cold stratification (Figure 7B). Previous studies have concluded that ERF TFs play vital roles in the plant chilling response and the transition from seed/bud dormancy to germination [35,67]. Cold-inducible *CBFs* increase plant chilling/freezing tolerance and are involved in the bud dormancy release of several perennials, including pear and peach [68,69]. The ERF TF genes *ABI4*, *DDF1*, and *EBE* play pivotal roles in seed dormancy and germination regulation [35]. Among them, *ABI4* plays multifaceted roles in hormone- and metabolite-related pathways, including ABA, GA, ETH, TAG, and soluble sugars and their cross talks [70]. Here, a homologous of *ABI4* (*Chr1.g57127*) was identified with down-regulated expression profiles during seed cold stratification, indicating its potential role in regulating the seed dormancy release of callery pear (Figure 8). Additionally, activator- and repressor-type *bHLH*, *bZIP*, *NAC*, *WRKY*, and *MYB* genes affect the seed chilling response and the seed dormancy status [35,71]. *ICE1*, a *bHLH* TF with a master role in modulating the plant cold response, physically interacts with a bZIP-type TF *ABI5* to fine-turn ABA signaling and suppress seed germination [72]. SA and ABA promote *HvWRKY38* expression and consequently suppress the GA-inducible α-amylase further during the seed dormancy process [73]. *DOG1*, a putative DNA-binding transcription factor and a key regulator of seed dormancy, physically interacts with two PP2C proteins (AHG1 and 3) to functionally block their essential roles in the release of seed dormancy [74,75]. The expression profile of *Chr15.g03683* (a homolog of *DOG1*) was rapidly increased from day 0 to day 20 of cold stratification and then decreased from day 20 to day 50 of cold stratification, and no obvious correlation was found between *DOG1* transcript levels and the potential of germination. The key regulators of pear seed dormancy release remain to be elucidated. According to our data, numerous TFs belonging to different families have diverse expression profiles in response to moisture chilling and might have regulatory functions in seed dormancy regulation (Figure 7).

## 4. Materials and Methods

### 4.1. Plant Materials and Treatments

Callery pear is an ornamental species of pear tree native to China and has been widely used as rootstock for grafting edible cultivars. Fruits of callery pear were harvested at the ripening stage from the garden of the College of Horticultural and Plant Protection, Yangzhou University. The seeds were separated from the fruits, washed to remove the pulp, and dried in the shade in ambient temperature. After 24 h of water uptake (imbibition), approximately 90 seeds per replicate (three replicates) of imbibed pear seeds were placed on petri dishes with wet filter paper and then held in cold storage (4 °C) for 50 days. The filter paper was moistened with water every 2–3 days. Seed samples were harvested at 0, 10, 20, 30, 40, and 50 days after cold stratification and divided into two bulks: 45 seeds were collected randomly to conduct a germination test, the other 45 seeds were immediately frozen in liquid nitrogen and stored at −80 °C for physiological determination and RNA sequencing.

To estimate the seed dormancy level, a germination experiment was performed with seeds sampled at discrete periods of cold stratification. Forty-five seeds per replicate were placed on petri dishes with moistened paper and cultured at 25 °C under 16 h of daylight (80 μmol m^−2^ s^−1^) and 70% relative humidity in a growth chamber. The germination percentages were determined after 10 days of culturation. The seeds were considered to have germinated when the radicle length was 5 mm.

### 4.2. Measurements of Hormone and Sugar Contents

According to the seed germination trial, seeds with 0, 20, and 50 days of stratification were used for sugar and hormone extraction and determination. Endogenous hormones, including ABA, GA3, and IAA, were extracted according to Wang et al. [76]. Hormonal quantification was carried out using high-performance liquid chromatography–electrospray ionization–mass spectrometry (HPLC-ESI-MS, Agilent 1200 UHPLC/6460 QQQ, Santa Clara, CA, USA). High-performance liquid chromatography was performed with an Agilent Zorbax XDB C18 column (150 mm × 2.1 mm × 3.5 μm). The mobile phases and gradient were as follows: mobile phase A, 0.1% (*v*/*v*) formic acid; mobile phase B, methanol. The flow rate was set to 0.3 mL/min. The gradient program was as follows: 60% A and 40% B for 1.5 min; 100% B for 6.5 min; and then 60% A and 40% B for 5 min. The content of ABA, GA3m and IAA was determined using the external standard method and is expressed as ng/g FW. The starch content was measured by the dinitrosalicylic acid (DNS) method described by Jeong et al. [77]; and soluble sugars, including sucrose, glucose, and fructose, were measured as described by Miao et al. [78].

### 4.3. RNA Extraction and Transcriptome Sequencing

For 9 seed samples, total RNAs were extracted by the ethyl ammonium bromide (CTAB) method as described by Zhang et al. [79]. RNA (1 μg) was used to generate an RNA-seq library with NEB Next^®^ UltraTM RNA Library Prep Kit for Illumina (NEB, Ipswich, MA, USA). After quality assessment, the libraries were sequenced on the Illumina Novaseq platform by Novagene (Beijing, China). The sequencing data (raw reads) of fastq format were firstly processed, and clean reads were filtered using FastQC to remove adapter, poly-N, and low-quality reads. The resulting clean data were mapped to the *P. betuleafolia* genome (https://ngdc.cncb.ac.cn/gwh/Assembly/647/show, accessed on 20 September 2020) with the Hisat2 program and then normalized into fragments per kilobase of the transcript sequence per millions (FPKM) values.

### 4.4. Differential Expression Analysis

The differences in expression between three stages were analyzed by DESeq with adjusted *p*-values < 0.05 corrected by Benjamini and Hochberg. Differentially expressed genes (DEGs) were defined according to the criteria: mean FPKM value for three biological replicates ≥1, absolute value of the log2 (fold change) ≥1, and a correlated *p*-value ≤ 0.01.

For all DEGs, we performed k-means clustering to show the expression patterns of genes with the same or similar expression behavior and divided the DEGs into eight clusters. In each cluster, the DEGs were annotated with the Kyoto Encyclopedia of Genes and Genomes (KEGG) database for functional enrichment analysis. Pathways with correlated *p*-values ≤ 0.01 were considered statistically significant.

### 4.5. Quantitative Real-Time PCR (qRT-PCR) Validation

To validate the accuracy of transcriptome profiling, the expressions of 12 DEGs related to phytohormone, sugar, and lipid pathways were evaluated by qRT-PCR, RNA extraction, first-strand cDNA synthesis, and qRT-PCR as per a previously described method [79]. After checking the quality and specificity of gene-specific primers, qRT-PCR reactions were carried out using the Bio-Rad CFX96 instrument (Bio-Rad, Hercules, CA, USA). Relative gene expression analysis was examined using the 2-ΔCt method with three biological replicates. The qRT-PCR primers are listed in Appendix A.

### 4.6. Statistical Analysis

The significance of hormone and sugar results was calculated according to Duncan’s test at *p* < 0.05 and indicated by different lowercase letters (a, b, c). Bar graphs were constructed using GraphPad Prism 7.0 scientific software (San Diego, CA, USA).

## 5. Conclusions

By combining the physiological and transcriptomic analyses, we constructed the regulatory network of seed dormancy release in callery pear. During the three crucial times of cold stratification, significant alterations in hormone contents and carbohydrate metabolism were observed and reflected the dormancy release process of pear seeds. Our comparative transcriptome studies uncover dozens of genes related to plant hormones and sugars that are significantly consistent with physiological changes. Genes related to ABA, GA, and IAA are identified as the main targets for the maintenance and release of pear seed dormancy. The mobilization of stored starch into soluble sugars is activated during the cold stratification of pear seeds. The genes related to carbohydrate metabolism as well as lipid metabolism are activated, providing essential energy sources and contributing to seed dormancy release. Furthermore, a plethora of TF genes exhibit dynamic expression patterns during the cold stratification of callery pear seeds and are thought to be crucial in seed dormancy release regulation. Consequently, the findings obtained in this study enrich the information on dormancy release regulation in pear seeds.

## Figures and Tables

**Figure 1 ijms-23-02186-f001:**
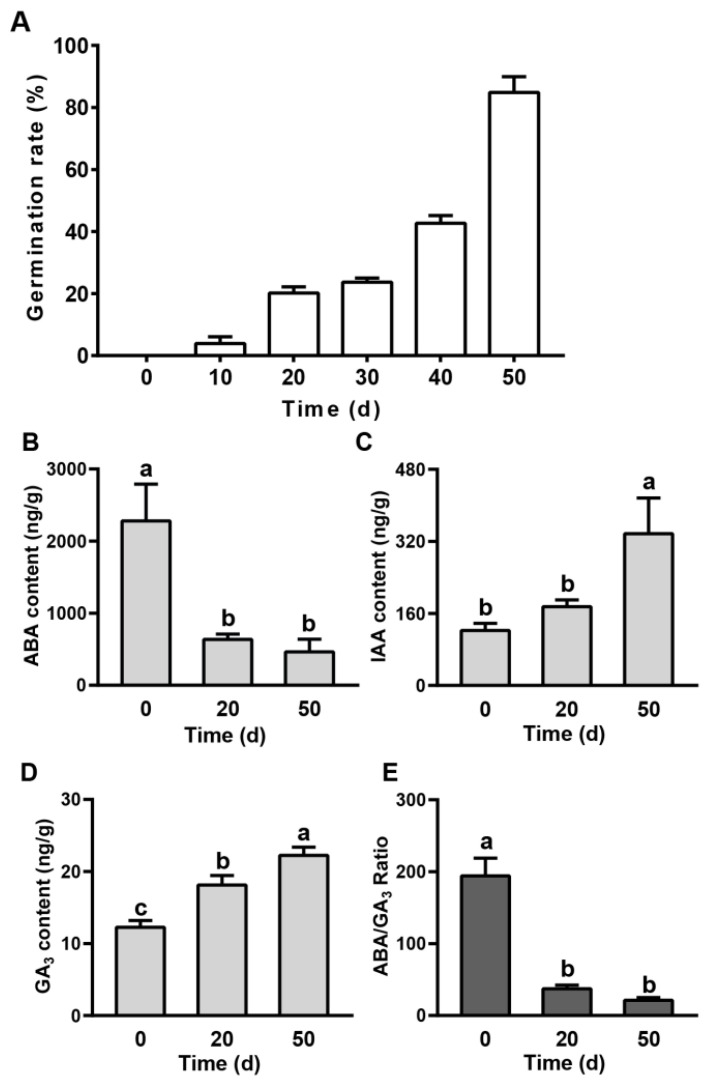
Characterization and measurement of phytohormones across the dormancy release stages of callery pear seeds. (**A**) Seed germination rate measured after discrete periods of cold stratification. Every 10 days after cold stratification, 90 seeds were placed in three petri dishes filled with wet filter paper and cultured in the growth incubator at a constant temperature of 25 °C and the germination rate was tested on day 10. (**B**) ABA, (**C**) GA3, (**D**) IAA, and (**E**) ABA/GA3 changes in seeds with different dormancy release statuses. Values are presented as the mean ± the standard error (SE) with three biological replicates. The different letters above the bars indicate the least significant differences between values with a confidence level of 95%.

**Figure 2 ijms-23-02186-f002:**
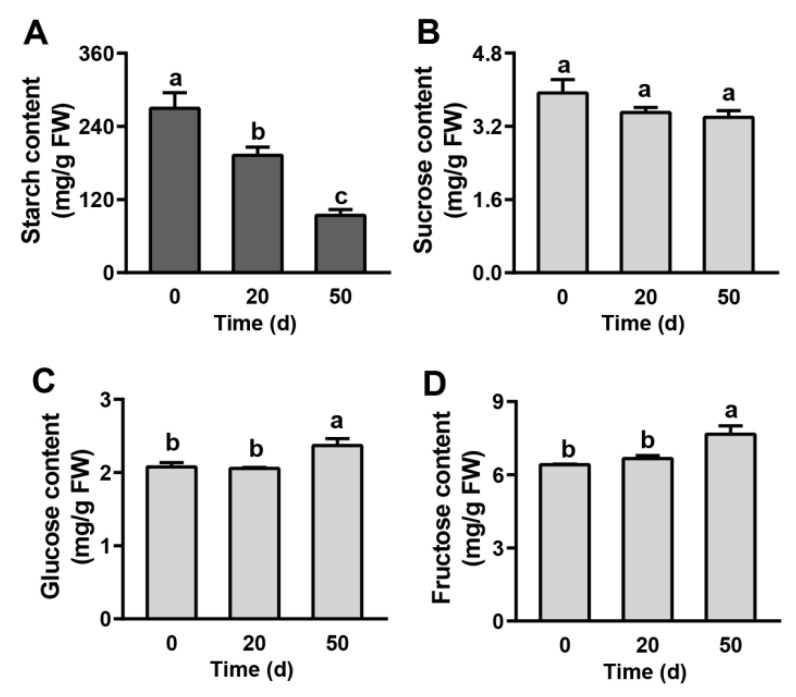
Measurement of carbohydrate contents across the dormancy release stages of callery pear seeds. (**A**) starch, (**B**) sucrose, (**C**) glucose, and (**D**) fructose. Values are presented as means ± SE with three biological replicates. The different letters above the bars indicate the least significant differences between values with a confidence level of 95%.

**Figure 3 ijms-23-02186-f003:**
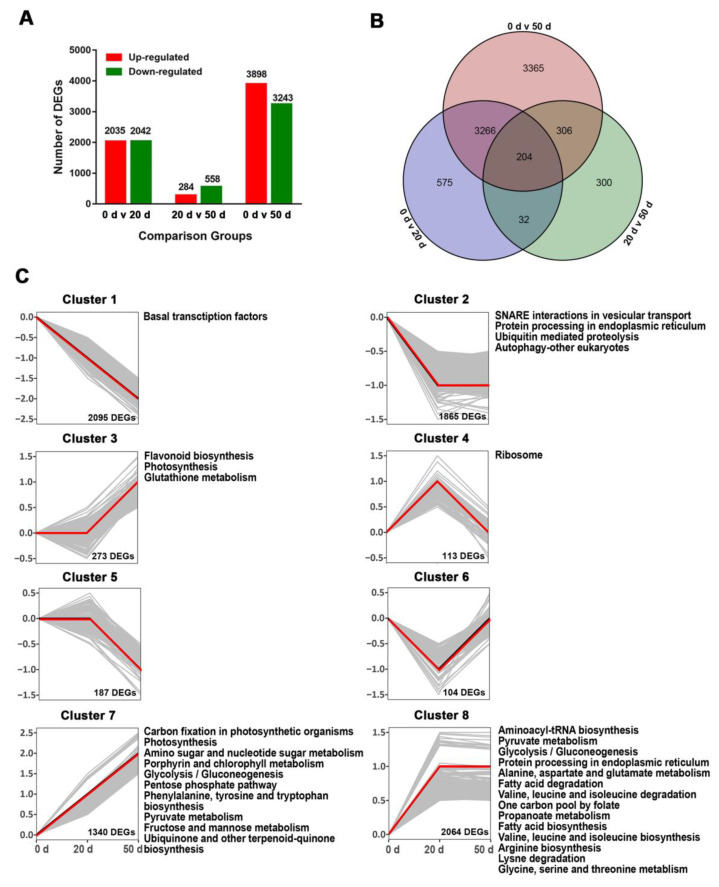
Identification and expression analysis of the differentially expressed genes (DEGs) involved in the dormancy release regulation of callery pear seeds. (**A**) Comparison of the number of DEGs among the three comparison groups and the number of up- and down-regulated DEGs in the individual comparison groups. (**B**) Venn diagram of DEGs among the three comparison groups. (**C**) Cluster diagrams of differentially expressed genes (DEGs) based on pairwise comparisons. *k*-means clustering of 8048 DEGs was performed based on Pearson correlation of gene expression profiles. The red line in each panel represents the mean pattern of expression of the DEGs for each cluster. The DEGs in each cluster were examined with KEGG enrichment analysis based on the criterion of a correlated *p*-value ≤ 0.01.

**Figure 4 ijms-23-02186-f004:**
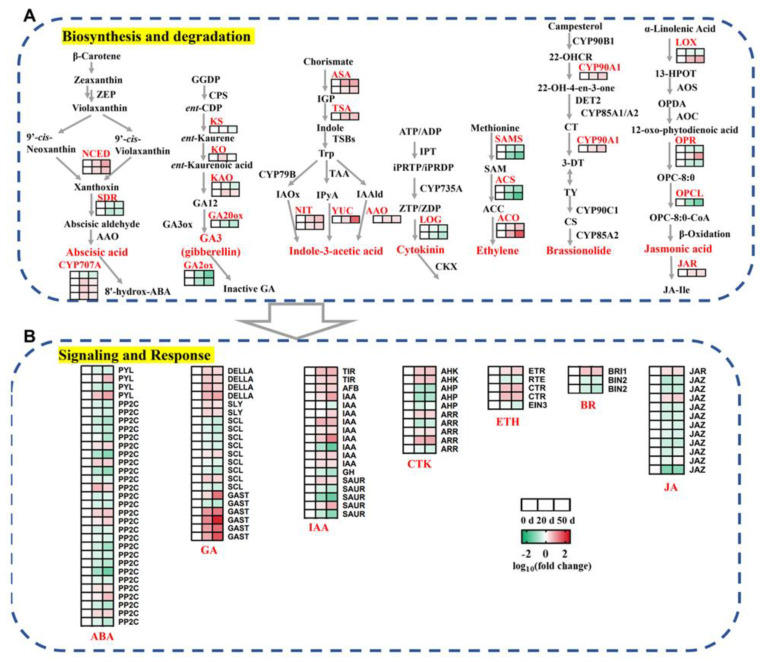
Regulatory model of DEGs involved in pathways related to plant hormone biosynthesis, degradation, signaling, and response (including ABA, GA, IAA, CTK, ETH, BR, and JA) in callery pear seeds during seed dormancy release regulation. (**A**) Expression heatmaps of the DEGs involved in plant hormone biosynthesis and degradation. (**B**) Expression heatmaps of the DEGs involved in plant hormone signaling and response. Gradient colors represent log_10_ (fold change) in gene expression at different times (0, 20, and 50 days of cold stratification of pear seeds); red represents high expression, whereas green represents low expression.

**Figure 5 ijms-23-02186-f005:**
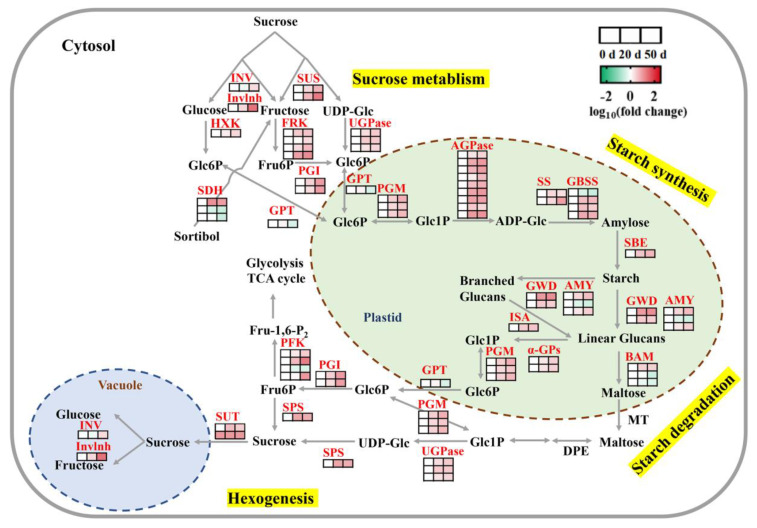
Regulatory model of DEGs involved in starch and sucrose metabolism pathways in callery pear seeds during dormancy release regulation. Expression heatmaps of the DEGs involved in sucrose metabolism, starch synthesis, starch degradation, and hexogenesis in the seeds are shown. Gradient colors represent log_10_ (fold change) in gene expression at different times (0, 20, and 50 days of cold stratification of pear seeds); red represents high expression, whereas green represents low expression.

**Figure 6 ijms-23-02186-f006:**
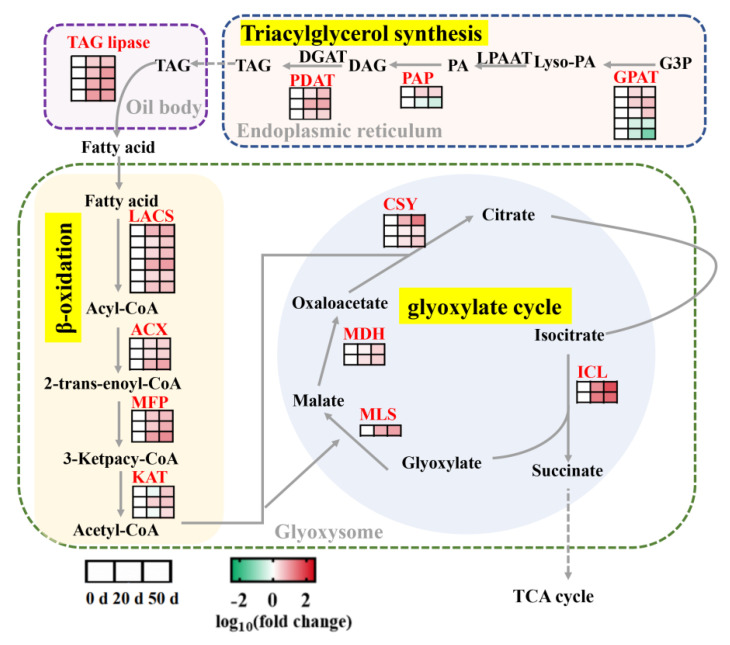
Regulatory model of DEGs involved in lipid synthesis and catabolism pathways in callery pear seeds during dormancy release regulation. Gradient colors represent log_10_ (fold change) in gene expression at different times (0, 20, and 50 days of cold stratification of pear seeds). Red represents high expression, whereas green represents low expression.

**Figure 7 ijms-23-02186-f007:**
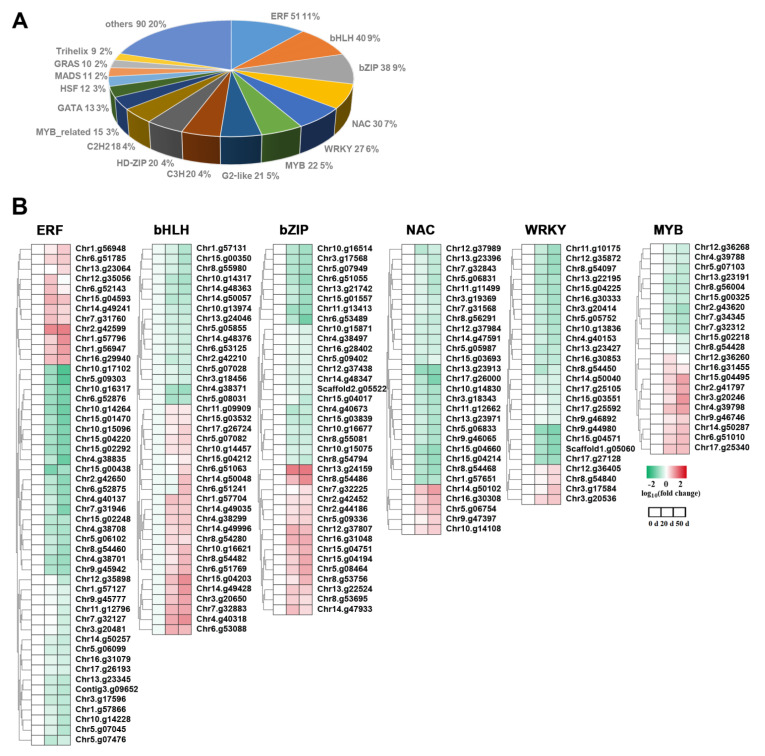
Transcription factors were differentially expressed in callery pear seeds during dormancy release regulation. (**A**) The number of differentially expressed transcription factors in different families. (**B**) Expression heatmaps of the DEGs encoding transcription factors. Gradient colors represent log_10_ (fold change) in gene expression at different times (0, 20, and 50 days of cold stratification of pear seeds). Red represents high expression, whereas blue represents low expression.

**Figure 8 ijms-23-02186-f008:**
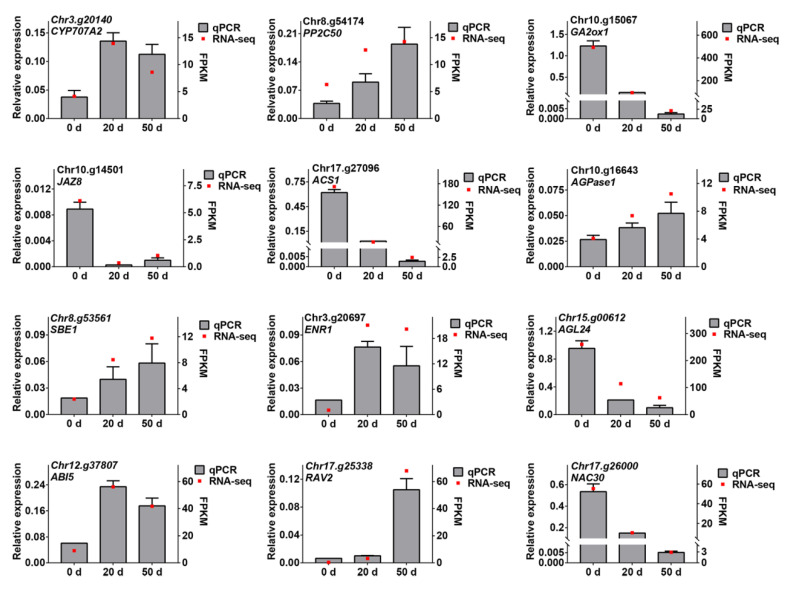
Quantitative qRT-PCR validation of DEGs. Twelve genes were selected for the quantitative qRT-PCR experiments. The relative expression levels of qRT-PCR were calculated using actin as a standard: *CYP707A2* (ABA 8′-hydroxylase), *PP2C50* (protein phosphatase 2C), *GA2ox1* (gibberellin 2 oxidase), *JAZ8* (jasmonate ZIM domain-containing protein), *ACS1* (1-aminocyclopropane-1-carboxylate synthase), *AGPase1* (ADP-glucose pyrophosphorylase), *SBE1* (starch branching enzyme), *MLS* (malate synthase), *AGL24*, *ABI4*, *RAV2*, and *NAC42*.

## Data Availability

The RNA-seq datasets generated during the current study have been deposited in the Gene Expression Omnibus (GEO) database with the project ID GSE192866, and other data supporting the results are included in this published article and its Appendix A files.

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
