# Peer review of "Transcriptome and Metabolite Conjoint Analysis Reveals the Seed Dormancy Release Process in Callery Pear"

_ijms, 2022, doi:10.3390/ijms23042186_

Round 1

Reviewer 1 Report

With the goal of understanding the molecular mechanisms that underlie seed dormancy release, Zhang et al. use RNA sequencing, and phytohormone and sugar content measurements to analyse transcriptome profiles at three different stages of cold stratification in callery pear seeds. The authors found a relationship between hormonal and gene expression changes for a substantial number of genes, including genes involved in hormone metabolism and signal transduction, several transcription factors and genes associated with sugar and lipid metabolisms. Overall, the authors found evidence of dormancy release in callery pear being a transcriptionally regulated process that involves complex molecular interactions.

The study advances our knowledge of key genes involved in seed dormancy release in economically important fruit trees, expectedly with possible impacts on horticultural approaches including the use of grafting, scions and rootstocks.

The paper is nicely written (but see comments below). The problem is clearly presented, and the experimental design seems to be appropriate. Results are detailed with nice figures and the discussion is comprehensive.

Minor comments:

The authors take samples at different times during cold stratification (lines 485-489) and therefore it would be more appropriate to use the term “time” rather than “stages” when presenting the data. The fact that seeds from the three selected times show different germination rates is not an a priori condition for categorizing those times into different stages.

Check that there is no misplacement of the different sections. Results appear in section 2, but M&M is presented in section 4 after the Discussion (section 3) but before the Conclusion (section 5).

Abstract/Introduction:

lines 27-28: “…might be related…” or “…might relate…”

line 58: “…in seeds,…”

line 63: delete the dot in “…and germination. [10,11].”

line 72: “…The expression levels of the ABI genes were significantly differentiated in pear seeds…” or “…The ABI genes were significantly differentially expressed in pear seeds…”

Results:

line 114: replace the semicolon for a dot.

line 191: in “P-value” the “p” without capital and italicized.

lines 193, 197, 255, 291 and so on: “…during seed dormancy…”

lines 264, 294 and so on: “…during seed cold stratification…” or “…during cold stratification of seeds…”

Discussion:

Perhaps dividing the discussion in 2-3 subsections with titles stressing the key results/conclusions at transcriptional, hormonal or sugar/lipid metabolic levels will help the reader to visualize the main findings.

Conclusion:

The conclusion is too general and does not focus on the results of the present study.

line 550: delete “comprehensive”, a regulatory network cannot be comprehensive, but the knowledge about it can be comprehensive. However, this is not the case here as the authors are making the first steps toward the molecular characterization of seed dormancy in pear.

Figures 3-8 are too small and the details on them cannot be read.

References:

check the style. Some references show the title in capital letters (e.g., 47, 53, 59).

Author Response

Response to Reviewer 1 Comments

Point 1: The authors take samples at different times during cold stratification (lines 485-489) and therefore it would be more appropriate to use the term “time” rather than “stages” when presenting the data. The fact that seeds from the three selected times show different germination rates is not an a priori condition for categorizing those times into different stages.

Response 1: Thank you for the suggestion. We have revised the paper and replaced S1, S2 and S3 with 0 d, 20 d and 50 d of cold stratification accordingly.

Point 2: Check that there is no misplacement of the different sections. Results appear in section 2, but M&M is presented in section 4 after the Discussion (section 3) but before the Conclusion (section 5).

Response 2: We have checked the place of the different sections according to the reviewers’ suggestion.

Point 3:

Abstract/Introduction:

lines 27-28: “…might be related…” or “…might relate…”

line 58: “…in seeds,…”

line 63: delete the dot in “…and germination. [10,11].”

line 72: “…The expression levels of the ABI genes were significantly differentiated in pear seeds…” or “…The ABI genes were significantly differentially expressed in pear seeds…”

Results:

line 114: replace the semicolon for a dot.

line 191: in “P-value” the “p” without capital and italicized.

lines 193, 197, 255, 291 and so on: “…during seed dormancy…”

lines 264, 294 and so on: “…during seed cold stratification…” or “…during cold stratification of seeds…”

 Response 3: We have revised the paper and improved the English grammar according to the suggestions.

Point 4:

Discussion:

Perhaps dividing the discussion in 2-3 subsections with titles stressing the key results/conclusions at transcriptional, hormonal or sugar/lipid metabolic levels will help the reader to visualize the main findings.

Response 4: Thank you for the suggestion. We have divided the discussion into 3 subsections with titles accordingly.

Point 5:

Conclusion:

The conclusion is too general and does not focus on the results of the present study.

line 550: delete “comprehensive”, a regulatory network cannot be comprehensive, but the knowledge about it can be comprehensive. However, this is not the case here as the authors are making the first steps toward the molecular characterization of seed dormancy in pear.

 Response 5: Thank you for the suggestions. We have largely revised the conclusion and the main implications of the results were summarized and added in the revised Conclusion section (Lines 554-566).

Point 6: Figures 3-8 are too small and the details on them cannot be read.

 Response 6:  The size and quality of figures 3-8 have been improved according to the reviewers’ suggestion.

Point 7:

References:

check the style. Some references show the title in capital letters (e.g., 47, 53, 59).

Response 7:  Sorry for the mistakes. We have checked and modified the references carefully.

Reviewer 2 Report

The manuscript by Zhang et al. is about a transcriptome and metabolite survey trying to disentangle the dormancy release process in Callery Pear. The topic is of high importance for the improvement of pear nursery stock to be used for the cultivation of Pyrus spp. However, the study presents some important flaws that need to be addressed before it can be considered for publication in IJMS. 

Firstly, the DOG1 gene, which is one of few genes associated with seed dormancy recognized by the seed physiology community, was not analyzed in the transcriptome data and was even not mentioned in the study. The transcript related to this gene should be searched in the transcriptome data and his expression pattern should be discussed taking into consideration ABA biosynthesis pathway.

Secondly, the introduction represents a very important section of the paper and it should provide the proper background in a clear sequence of information. At the moment this section is very confused, es.: lanes 50-52 needs to be moved upward (on lances 44-46) when the dormancy of the species is presented.

Thirdly, the treatment for the assessment of the dormancy state is confusing. In Lane 108 the authors stated, “after 30 d of cold stratification seed germination was rapidly increased.”. This sentence is not confirmed by the graph in which a significant increase can be observed only after 50d. Moreover, the choice of the kind of graph for Figure1A t does not seem the most appropriate. Namely, the different times derives from different experiments and it is not a time-course analysis as the graph looks like. It should be graphed as a bar graph or in any case as independent experiments. Finally, at lane 106 the authors stated that for this experiment seeds were “cultured at room temperature for 10d”. Incubation at room temperature is, definitely, not a reliable temperature parameter. This parameter is of crucial importance and a controlled temperature incubation should be used.

Minor flaws:

Results:

· Figures 4, 5 and 6: The expression range scale should be specified as fold change (“high” and “low expression” are too generic).

· Please revise the sentence at lanes 340-341

Discussion:

· WRKY upregulation at the S1 stage seems clear evidence of association of these TFs with dormancy maintenance and so should be further investigated and discussed.

· Lanes 425 – 427: in the present study the authors reported data on degradation only related to starch and no direct observations are provided for fatty acids or protein degradation. So the sentence should be modified according to what is reported in this study.

M&M:

· Lanes 507-508: more specifications about the RNA extraction protocol should be provided. There a plenty of different versions of the CTAB protocol. The authors should refer to a proper study or provide all the passages.

References:

· Finkelstein 1994 is a very old reference please update the reference with a more recent study on abi3 gene

Author Response

Response to Reviewer 2 Comments

Point 1: Firstly, the DOG1 gene, which is one of few genes associated with seed dormancy recognized by the seed physiology community, was not analyzed in the transcriptome data and was even not mentioned in the study. The transcript related to this gene should be searched in the transcriptome data and his expression pattern should be discussed taking into consideration ABA biosynthesis pathway.

 Response 1: According to the reviewers’ suggestion, we reanalyzed the differentially expressed gene list and obtained Chr15.g03683 that was homologous to DOG1 (Line 318). The expression profiles of DOG1 were discussed taking into consideration ABA signaling transduction pathway (Lines 474-480).

Point 2: Secondly, the introduction represents a very important section of the paper and it should provide the proper background in a clear sequence of information. At the moment this section is very confused, es.: lanes 50-52 needs to be moved upward (on lances 44-46) when the dormancy of the species is presented.

 Response 2: Thank you for the suggestion. We have revised the introduction section accordingly.

Point 3: Thirdly, the treatment for the assessment of the dormancy state is confusing. In Lane 108 the authors stated, “after 30 d of cold stratification seed germination was rapidly increased.”. This sentence is not confirmed by the graph in which a significant increase can be observed only after 50d. Moreover, the choice of the kind of graph for Figure1A t does not seem the most appropriate. Namely, the different times derives from different experiments and it is not a time-course analysis as the graph looks like. It should be graphed as a bar graph or in any case as independent experiments. Finally, at lane 106 the authors stated that for this experiment seeds were “cultured at room temperature for 10d”. Incubation at room temperature is, definitely, not a reliable temperature parameter. This parameter is of crucial importance and a controlled temperature incubation should be used.

 Response 3: Thank you for the suggestions. We have revised the description of dormancy state during seed cold stratification (Line109) and specified the culture condition of pear seed germination experiment (Line106). Figure1A has been reorganized to bar chart.

Point 4:

Results:

  • Figures 4, 5 and 6: The expression range scale should be specified as fold change (“high” and “low expression” are too generic).

 Response 4: The expression range scale of Figures 4-7 has been revised and specified as fold change according to the reviewers’ suggestion.

Point 5: Please revise the sentence at lanes 340-341

 Response 5: The sentence has been revised.

Point 6: Discussion:

WRKY upregulation at the S1 stage seems clear evidence of association of these TFs with dormancy maintenance and so should be further investigated and discussed.

 Response 6:·Sorry for the misunderstanding, the gradient color of differentially expressed WRKY heatmap in the previous Figure 7 was different from the other differentially expressed TF heatmaps and gave a false view. The mean FPKMs of differentially expressed WRKY genes were not significantly higher in 0 d of cold stratification compared with the other differentially expressed TFs. In the present Figure 7, the expression range scale has been revised and specified as fold change instead of FPKM.

Point 7: Lanes 425 – 427: in the present study the authors reported data on degradation only related to starch and no direct observations are provided for fatty acids or protein degradation. So the sentence should be modified according to what is reported in this study.

 Response 7: Thank you for the suggestion. We have modified the sentence accordingly.

Point 8:

M&M:

Lanes 507-508: more specifications about the RNA extraction protocol should be provided. There a plenty of different versions of the CTAB protocol. The authors should refer to a proper study or provide all the passages.

 Response 8: Thank you for the suggestion. The quoted reference of RNA extraction protocol has been added in Materials and Methods section (Line 521).

Point 9: References:

  • Finkelstein 1994 is a very old reference please update the reference with a more recent study on abi3 gene

 Response 9: Thank you for the suggestion. We have updated the reference of ABI3 accordingly (Lines 620-621).

Reviewer 3 Report

The goal of this manuscript is to present and discuss the sisgnificant changes in the transcriptome and metabolome during dormancy releasing of Pyrus calleryana Decne seeds. For this purpose the authors had various approaches by analyzing trnascriptome profiles using RNA-seq data, hormone level (ABA, GA3, IAA) using HPLC-ESI-MS, and starch and soluble sugar level. They conclude that (1) ABA, GA and IAA are important for the maintenance and release of Pyrus calleryana seed dormancy, (2) starch and sucrose, and lipids metabolism are significantly upregulated during dormancy release progression.
All parts of the manuscript is interesting and clearly summarize new data valuable for the research community. The author has done an good job at describing all topics.

GENERAL COMMENTS:
TITLE
The paper title is well stated, it is informative and concise. 

ABSTRACT, INTRODUCTION
Abstract and Introduction were well written.

MATERIAL AND METHODS
Material and research methods are presented appropriately and clearly. Experimental setup and the description in the methods section are well structured, and the statistical analysis is done alright. In spite of that I have a few objections against its present form. These are listed below:

4.2. Measurements of hormone and sugar contents
- please indicate the conditions of the analysis by means of HPLC;
4.3. RNA extraction and transcriptome sequencing
- where is the RNA-seq data deposited?

RESULTS
The results obtained in this study are interesting. Results presented correctly. 

DISCUSSION
In general, the discussion of results is correct and sufficient.

LITERATURE
The items of literature included in the paper are rather sufficient and adequate to the subject of the paper.

The text of the manusctipt is not formatted correctly yet.
Please verify the correctness of the literature and make a linguistic correction of the text by native speaker.

Author Response

Response to Reviewer 3 Comments

Point 1: GENERAL COMMENTS:
TITLE
The paper title is well stated, it is informative and concise. 

ABSTRACT, INTRODUCTION
Abstract and Introduction were well written.

 Response 1: Thank you for the affirmation about the title, abstract and introduction.

Point 2: MATERIAL AND METHODS
Material and research methods are presented appropriately and clearly. Experimental setup and the description in the methods section are well structured, and the statistical analysis is done alright. In spite of that I have a few objections against its present form. These are listed below:

4.2. Measurements of hormone and sugar contents
- please indicate the conditions of the analysis by means of HPLC;
4.3. RNA extraction and transcriptome sequencing
- where is the RNA-seq data deposited?

 Response 2: Thank you for the suggestions. We have supplied the conditions of HPLC for hormone analysis (Lines 507-515). The RNA-seq data deposition was described in Data Availability Statement section (Lines 585-588) and the datasets were deposited in the Gene Expression Omnibus (GEO) database with the project ID GSE192866.

Point 3:

RESULTS
The results obtained in this study are interesting. Results presented correctly. 

DISCUSSION
In general, the discussion of results is correct and sufficient.

LITERATURE
The items of literature included in the paper are rather sufficient and adequate to the subject of the paper.

 Response 3: Thank you for the affirmation about the results, disscussion and reference section.

Point 4:

The text of the manusctipt is not formatted correctly yet.
Please verify the correctness of the literature and make a linguistic correction of the text by native speaker.

 Response 4: We have largely revised the paper and improved the English grammar. Moreover, the English language was also improved by native English speaking.

Round 2

Reviewer 2 Report

the new version of the manuscript responds to the raised flaws.